# Updating Chimpanzee Nesting Data at Mount Assirik (Niokolo Koba National Park, Senegal): Implications for Conservation

**DOI:** 10.3390/ani14040553

**Published:** 2024-02-07

**Authors:** Yaya Hamady Ndiaye, Papa Ibnou Ndiaye, Stacy Marie Lindshield, Jill Daphne Pruetz

**Affiliations:** 1Département de Biologie Animale, Faculté des Sciences et Techniques, Université Cheikh Anta Diop, Dakar 5005, Senegal; yayahamadyndiaye@gmail.com; 2Observatoire International Homme-Milieux Téssékéré, IRL3189 “Environnement, Santé, Sociétés”, Université Cheikh Anta Diop, Dakar 5005, Senegal; 3Department of Anthropology, Purdue University, West Lafayette, IN 47907, USA; slindshi@purdue.edu; 4Department of Anthropology, Texas State University, San Marcos, TX 78666, USA; pruetz@txstate.edu

**Keywords:** chimpanzee, nesting, Niokolo Koba National Park, Senegal

## Abstract

**Simple Summary:**

The aim of protected areas is to conserve nature. However, the wild animal populations they shelter are not immune to extinction. It is, therefore, important for researchers to provide regular information on the ecology of these populations in order to guide management actions and enhance knowledge. From studies based on chimpanzee nests, we revealed that chimpanzees in Niokolo Koba National Park nest more in closed habitats than in open ones, and that certain plant species are preferentially chosen for nest construction. *Diospyros mespiliformis* and *Pterocarpus erinaceus* are the trees most used by chimpanzees for bearing their nests. In addition, we found that chimpanzee nest height and nest life were influenced by the environmental factors. We conclude that the preservation of nesting sites of chimpanzees is crucial for the conservation of this great ape, considered to be on the border of extinction in Senegal.

**Abstract:**

The Niokolo Koba National Park (NKNP) is the largest protected area in Senegal and lies at the northern limit of the chimpanzee’s range in West Africa. Recent information on nesting behavior and factors influencing nesting behavior is available for several sites outside NKNP. However, the information available for NKNP is obsolete. Considering that the adequate management of chimpanzee populations cannot be achieved without strong scientific knowledge, it is essential to update data on chimpanzee, *Pan troglodytes verus*, nesting behavior in NKNP. For this reason, we surveyed their habitat in Mt. Assirik and recorded 626 chimpanzee nests. The results of the study showed that chimpanzees nest more often in closed-canopy habitats such as gallery forests. The average nest height observed in this study was 8.07 ± 0.36 m, varying between 2 and 20 m, which is well below the heights described in most sites where chimpanzees cohabit with large carnivores. Botanical surveys confirmed that chimpanzees select tree species bearing their nests. In Assirik, 12 of the 37 tree species bearing nests are the most used. The nest decay rate (or the time it takes for a nest to go from the fresh to the skeletal stage) at Assirik averaged 68.8 ± 5.8 days.

## 1. Introduction

The chimpanzee (*Pan troglodytes*), like all of the great apes, is classified as “endangered” throughout its wild range on the International Union for Conservation of Nature (IUCN) Red List [1,2]. Recent population status assessments show that West Africa is home to one of the most threatened populations [3], classified as critically endangered [4] and thought to be on the brink of extinction in four of their nine range countries (Burkina Faso, Ghana, Guinea-Bissau and Senegal) [5]. There is, therefore, an urgent need to intensify research on chimpanzees in West Africa in order to better protect their population from extinction.

In Senegal, the distribution of chimpanzees is limited to the southeast of the country, in the Kedougou region and part of Kolda region, within the Niokolo Koba National Park (NKNP). Kedougou contains a large part of the Niokolo Koba National Park, which represents one of the last bastions of large wild mammals of West Africa [6] in this hot and dry savanna ecosystem, one of the last refuges for the chimpanzee in West Africa. The presence of this great ape in this area has contributed greatly to the enhancement of NKNP and also to the amplification of research in and around the park [7,8,9,10,11,12,13,14,15]. However, over the last few decades, this natural environment has been subjected to numerous natural and anthropogenic hazards (poaching, gold mining, cattle grazing, among others) [16], which have hindered the proper functioning of its ecosystems and led, in 2007, to its inclusion on UNESCO’s List of Word Heritage In Danger [17]. These constraints have led to a decline in the numbers of numerous animal species, such as the lion (*Panthera leo*), leopard (*Panthera pardus*), wild dog (*Lycaon pictus*), roan antelope (*Hippotragus equinus*), African buffalo (*Syncerus caffer*) and African elephant (*Loxodonta africana*) [12,18,19,20].

The current situation of degradation of natural resources is more than alarming on the periphery of the park, and is characterized by the intensive exploitation of flora and fauna, to which is added the competition of local residents for agricultural and silvopastoral land [21,22]. In addition, the establishment of large-scale mining operations and the phenomenon of transhumance [15,16] has even more serious consequences for biodiversity, including deforestation and habitat fragmentation.

These factors have multiple negative consequences for natural resources, leading to a sharp decline in the productive potential of natural resources, as well as degradation of the vegetation cover. For a species such as the chimpanzee, which lives in the unusual climatic conditions of a hot, dry tropical savanna [14,23] and is essentially a frugivore [16,23,24,25], the destruction of its habitat is a serious threat to the future of the chimpanzee in Senegal.

Although several studies on chimpanzees in the NKNP have resulted in numerous publications about their geographical distribution [26,27], social behavior [28], diet [25,29,30,31], ecology [32,33,34], anti-predator behavior [9,35], nesting [27,36,37], tool use [38], population demographics [39], and sympatric species [12,15], we need to improve our understanding of the different aspects of their behavior and ecology to support and enhance current population management strategies. For primatologists and wildlife managers, it is clear that the study of great ape nesting is very important because it is of great use in formulating effective conservation action plans [14]. Nest surveys make it possible to estimate population size, determine habitat quality [40], and identify preferences in terms of nest trees [41].

The nests consist of vegetative structures that can remain visible for weeks or months according to the nest decay rate, which depends on the site type, season, tree species bearing the nest and sun exposure. Weaned individuals build nests in which they sleep every night or sometimes rest during the day. Nest site selection is related to environmental factors, such as predator avoidance, human hunting pressure, climatic conditions, habitat types, tree species and the availability of ripe fruits, among others [37]. Direct observation of chimpanzees during field work is rare. Thus, many studies about great apes are focused on nests. However, variations in nesting behavior according to many factors implies the need to multiply studies for a better understanding of the ecology and behavior of chimpanzees in nature and better precision in the case of population estimations. Previous studies have shown that ape nest construction and related aspects of nesting behavior are closely linked to local ecological factors. For example, chimpanzee nest height depends on the presence of potential predators [9,35,40] and human hunting pressure [42]. In addition, the selection of nesting habitats is influenced by climatic conditions and vegetation density [10], and the rate of decay of chimpanzee nests is thought to depend on the season, the habitat, and the species of tree used to support the nests [40,41,43]. Researchers, therefore, need to provide local information on the nesting behavior of these great apes. Most of this information is available for many chimpanzee study sites in Senegal [9,10,11,12,13,14,16,37,41,44,45]. However, those of the Assirik group are outdated or insufficient [45], and good conservation management of this species depends above all on knowledge of the ecology and status of the populations. Consequently, the acquisition of data on the nesting behavior of chimpanzees in NKNP is necessary to improve the conservation and management of the species. Our study is part of this extension, in which we show the distribution of nests according to habitat type, nest height, tree species used for nest construction, and nest decay rate. Furthermore, we analyze environmental factors that are associated with nest characteristics. Ultimately, this updated information will help to protect and manage the chimpanzee in Senegal.

## 2. Materials and Methods

### 2.1. Study Site

Our study took place in the Niokolo Koba National Park in the Kédougou region of southeastern Senegal (Figure 1).

The Assirik chimpanzees are located in the eastern part of the park [9,34]. They were the first community to have been the subject of indirect long-term chimpanzee research in Senegal, between 1976 and 1979 [34,46]. The environment at Assirik can be described as an open Sudano-Guinean savannah formation [47] composed of heterogeneous vegetation types organized as grassland, dense tree savannah, shrub savannah, bamboo forests and gallery forests along watercourses, dominated by certain plant species such as *Saba senegalensis*, *Combretum tormentosum*, *Oncoba spinosa*, *Ficus umbellata* and *Diospyros mespiliformes* [48]. However, this vegetation structure has undergone major changes in recent decades with an extension of grasslands and shrub savannahs in favor of tree savannahs and gallery forests [49]. The transition from habitats with relatively tall trees to habitats with smaller trees could influence the nesting behavior of chimpanzees. The climate is hot and dry [50], with a long dry period lasting seven (7) months and a rainy season lasting around five (5) months between May and September. Average annual rainfall varies from 900 to 1200 mm, with an average annual temperature of 27.82 °C that ranges from 22 to 35 °C [16]. The terrain in Assirik is rugged, consisting of plateaus and valleys with gallery forests and hills, culminating in Mount Assirik at 311 m above sea level [48].

### 2.2. Data Collection and Analysis

The data collected during this study were recorded in an Excel worksheet, and statistical tests were carried out using R software (version number 4.3.2) [51].

#### 2.2.1. Distribution of Nests

Chimpanzee nest data were collected during several missions between December 2020 and December 2022. The Assirik chimpanzee home range is vast and difficult to access, encounters with chimpanzees are rare, and sightings of newly constructed nests are infrequent. Therefore, we began in December 2020 and March 2021 by determining the distribution of nests in the study area, walking reconnaissance previously defined following information and preliminary surveys by the Assirik Chimpanzee Research team. We covered the entire study area, with several reconnaissance walks oriented more towards the gallery forests, the preferred chimpanzee nesting habitat in NKNP [9,32,37]. When chimpanzee nests were encountered, the date, time, geographical coordinates, nesting tree species, habitat type and age class were recorded for each nest. The age classes were determined following the classification of Tutin and Fernandez [52]. According to these authors, the fresh nest is characterized by the presence of copious green and moist leaves, with, in most cases, fresh feces or urine on the ground beneath the nests; the recent nest is composed of withered, dried green leaves; the old nest is a mixture of dried green and brown leaves, or entirely brown leaves; the nest skeleton is mostly composed of the structural tree branches, but sometimes the remnants of brown leaves remain. We used QGIS 3.28 software to map the distribution of chimpanzee nests recorded during the study.

In addition, we assessed the influence of season on the distribution of nests in different habitat types. To do this, we calculated the proportion of nests in the different vegetation types as a function of season, and performed the chi-square test to determine the relationship that exists [11]. Given the difficulty of accurately determining the construction periods of old and skeletal nests, only fresh and recent nests recorded during the course of this study were used in this analysis.

#### 2.2.2. Estimation of Nest Decay Rate

The standing-crop nest count (SCNC) method for measuring the decay rate of chimpanzee nests is, arguably, the most robust approach for reliably determining nest lifespan [40,42,53]. However, applying this method to the savannah environments of Senegal neglects the effect of seasonal variations on nest decay rate in these ecosystems, as the estimated decay time does not take into account the impact of changing climatic conditions on decay rate [54]. For this reason, we used the marked-nest count (MNC) method, whereby nests were marked and tracked for a one-year period to measure the decay process. In this way, the estimated decay rate more effectively accounts for seasonal variance.

We monitored three (3) different groups of fresh nests (*n* = 166) marked at different times of the year: group 1 (G1, *n* = 61), made up of nests marked in March and degraded in June, the monitoring interval corresponding to the dry season until the start of the rainy season; in group 2 (G2, *n* = 36), the nests were marked in April and rotted in June, a period corresponding to the transition from the dry season to the months of the first rains in Kedougou; in group 3 (G3, *n* = 69), nests marked at the start of the rainy season (June) deteriorated in July, when it regularly rained. The team was made up of experienced observers able to recognize chimpanzee nests and the different stages of decomposition. Because of the rugged terrain and particularly the location of nests in the gallery forests at Assirik, we used the reconnaissance walk method [55] to record chimpanzee nests. Nest locations were marked so that old and new nests could be differentiated during subsequent surveys. The nests were revisited every month until the leaves had completely disappeared, leaving only the structural branches. In this way, the lifespan of each nest was accurately measured. We then used R software to calculate the average lifespan of the nests using a *t*-test (independent test) with a 95% confidence interval.

To determine the effect of environmental factors on nest decomposition, we carried out a logistic regression using the automated “stepwise” approach [43] in R. This technique works in such a way as to retain the best predictors of all of the variables included in the model. This selection of predictors is determined solely by AIC statistical criteria. We considered “nest lifespan” as the phenomenon to be explained and the variables “species of trees supporting nests”, “nest height” and “season” as explanatory factors. We excluded the habitat type variable from the regression because the vast majority of fresh nests were found in gallery forests.

#### 2.2.3. Tree Selection and Nest Height

In order to assess the botanical composition of tree species in the study area and the selection of plant species for nest construction, we carried out vegetation surveys at nesting sites and along reconnaissance paths in December 2022 (Figure 2). A total of 66 quadrats measuring 25 m × 25 m, spaced 250 m apart, were randomly placed along the reconnaissance paths where chimpanzee nests had been observed. In each quadrat, we identified and recorded all trees with a diameter at breast height (dbh) ≥ 10 cm. The relative abundance of each species was calculated, and we tested for a preference of nesting tree species by comparing the proportion of nests for each tree species against their relative abundance in the area [14,37]. In addition, we ran a linear regression in R software to check the preference of nesting trees.

During this survey, we recorded additional chimpanzee nests (*n* = 318) to determine the nest height of chimpanzees in the study area. To avoid counting a nest several times, we only recorded fresh and recent nests when the quadrats passed through areas already surveyed, and nests of all age classes in newly surveyed areas. The heights of these nests and their support trees were measured using a Tracker 670 laser rangefinder.

We calculate the average nest height and support trees using a *t*-test (independent test) with a 95% confidence interval and to assess the relationship between nest heights and tree heights using regression and correlation analyses.

## 3. Results

During surveys, monitoring and botanical surveys, we covered a total distance of 73 km. We recorded a total of 626 chimpanzee nests built on 259 trees, representing 31 genera and 17 families of 37 different tree species (Figure 3). Skeletal nests were recorded most often (41.85%, *n* = 262), followed by fresh (33.07%, *n* = 207), old (19.65%, *n* = 123), and recent (5.43%, *n* = 34) nests.

### 3.1. Distribution of Nests According to Habitat

The results revealed a significant difference (Kruskal–Wallis chi-squared = 10.861; df = 3; *p* < 0.05) between the rates of nest construction in the different habitat types. The great majority of nests (66.61%) were found in gallery forests, followed by woodland (14.7%), grassland (12.46%) and bamboo woodland (6.23%) (Figure 4). Chimpanzees tended to nest in tree-dominated habitats such as gallery forests and woodland, which together accounted for more than 80% of observations.

In the construction of nests in the types of vegetation, a significant difference was observed between seasons (Chi-square = 34.323, df = 2, *p* < 0.0001). Chimpanzees nested extensively in gallery forests during the dry season (*n* = 115, 84.56%). On the other hand, in the rainy season, there was an increase in nests in woodland and grassland, while the frequency of nesting decreased in gallery forests (Table 1).

### 3.2. Nest Height

Based on measurements of the height of 318 night nests, the average nest height of chimpanzees at Assirik was 8.07 ± 0.36 m. The height at which nests were built varied between 2 and 20 m above ground level, and chimpanzees nested at heights of between 5 and 15 m in 87% of cases, with more than 61% of these nests being between 5 and 10 m (Figure 5). In less than 5% of cases, the nests were more than 15 m above the ground. There was no significant difference in nest height according to habitat type (Kruskal–Wallis chi-squared = 2.4263, df = 3, *p* = 0.4887). Chimpanzees nested at approximately equal heights across the study site (Figure 6). However, nest heights between 15 and 20 m were most often observed in the forest galleries where the tallest trees were found.

In contrast to the habitat types, a statistically significant relationship (df = 316, *p* < 0.001) was found between nest height and support tree height. The average height of the nest trees was 9.82 ± 0.4 m. Linear regression between the two heights showed that they were highly correlated (r = 0.71, *n* = 318), so that nest height increased as a function of support tree height (Figure 7).

### 3.3. Choice of Plant Species for Nest Construction

Within the 66 quadrats, we identified a total of 1815 trees (dbh ≥ 10 cm) belonging to 64 different species. Of these 64 tree species, chimpanzees used at least 37 for nest construction, and 74.6% of these nests were constructed in eight species, including *Diospyros mespiliformis* (*n* = 119, 19.01%), *Pterocarpus erinaceus* (*n* = 104, 16.61%), *Malacantha alnifolia* (*n* = 58, 9.27%), *Ficus sur* (*n* = 43, 6.87%), *Celtis intigrifolia* (*n* = 39, 6.23%), *Afzelia africana* (*n* = 39, 6.23%), *Combretum collinum* (*n* = 36, 5.75%), and *Hexalobus monopetalus* (*n* = 29, 4.63%) (Figure 8). Relative to their abundance in the study area, Figure 8 shows that in addition to these eight species, chimpanzees showed a preference for nesting in *Pachystela pobeguiniana*, *Hannoa undulata*, *Khaya senegalensis* and *Cola cordifolia*. However, regression analysis showed that even if there was a preference for certain trees, the choice of tree species for nesting was partly linked to their abundance in the study area (r = 0.57, df = 62, *p* < 0.001) (Figure 9). Species such as *Diospyros mespiliformis* and *Pterocarpus erinaceus*, which are abundant in the site, were the most commonly used for nest construction.

### 3.4. Determination of Nest Decay Rate and Factors Influencing It

The average nest decay rate was 68.8 ± 5.8 days. The best model retained by the logistic regression retained the variables: species of trees used for nesting and season as the most important factors affecting the rate of nest decay (lifespan ~ Nest_trees + Group, Step: AIC = 1206.96). Nest height did not have a large effect on nest lifespan, as its presence in the starting model gave a higher AIC (lifespan ~ Nest_trees + Nest_height + Group, Start: AIC = 1208.31). However, the mean nest decay time was longer for nests built at heights of 10–20 m (87.2 days) than for those built at heights of 0–10 m (62.2 days).

Since logistic regression indicated tree species and season as the factors that best explained nest decay, we thus directly analyzed the impact of these two factors on decay rate. There was a significant difference (Kruskal–Wallis chi-squared = 133.44, df = 18, *p*-value < 2.2 × 10^−16^) in mean nest decay time between the different tree species used for nest construction. The average nest lifespan per species varied between 30 and 109 days. Plant species such as *Hannoa undulata*, *Pachystela pobeguiniana*, *Malacantha alnifolia*, *Celtis intigrifolia* and *Ficus sur* had high mean nest decomposition times, whereas they were relatively low for *Afzelia Africana*, *Lannea microcarpa*, *Pterocarpus erinaceus* and *Khaya senegalensis* (Table 2). With the exception of *Lannea microcarpa*, the coefficients for the tree species were positive, which suggested that the physical characteristics of certain trees increase the nest lifespan.

A significant difference (*p* < 0.001) was observed within the three groups of nests monitored at different periods. Nests built during the dry season took much longer to degrade and had an average lifespan 1.7 to 3 times longer than nests monitored during periods of lower rainfall and nests built during periods of high rainfall (Table 3).

Figure 10, which plots the number of decaying nests against time, shows that a large proportion of nests disintegrated after 60 days. Between the sixtieth and ninetieth day, few nests had degraded, corresponding to the remaining 9.8% of G2 nests. As for the nests that degraded after 90 days, they were made up exclusively of nests built in the dry season (G1).

## 4. Discussion

Chimpanzees in NKNP have shown a preference for nesting in closed habitats, particularly in gallery forests. The choice of gallery forest for nesting was reported by Baldwin [32] and Baldwin et al. [36] and later by Pruetz et al. [9], who suggested that this preference at Assirik was motivated by the safety of nesting in these closed habitats, where the trees are generally taller, allowing the chimpanzees to escape from sympatric predators more easily [12]. However, the choice of closed habitats could also be explained by the availability of water and food throughout the year [10]. In addition, in this type of habitat, tree leaves, which are one of the most important materials for nest construction, are available throughout the year, whereas in other types of habitats, many trees lose their leaves in the dry season [11]. The results in Table 1 show a significant increase in nest construction in woodland and grassland during this period, and conversely a reduction in gallery forests. These results are similar to those of [11], who showed that chimpanzee nesting in Senegal during the rainy season was directed towards gallery forest edges and high plateaus, perhaps due to the lack of trees suitable for nest construction in flooded gallery forests.

Chimpanzees nested at an average height of 8.07 ± 0.36 m, and most of the nests (87%) observed were between 5 and 15 m high in all types of habitat, even in gallery forest where taller trees are available. The average nest height at Assirik (this study) was similar to those obtained at some study sites in Senegal (Fongoli, Bagnomba and the Antenna zone of NKNP) [9,36,40], but lower than at other sites such as Diaguiri and Bantankiline [14,45] (Table 4). However, in a previous study at Assirik, Pruetz et al. [9] obtained 13.55 m as the average nest height. The strong correlation between nest height and nesting tree height indicates that chimpanzees at Assirik do not seek to sleep in the tallest available trees. It could be that the preference to build nests within a particular height range could be the underlying cause of the observed decrease in average nest height over time at Assirik.

The average nest height in our study was considerably lower than at most sites where they cohabit with a large number of large carnivores (Issa, Tanzania; Sapo, Liberia; Taï, Côte d’Ivoire; Semliki, Uganda) (Table 5). Alternatively, the reduction in the average nest height could be the result of changes in predation pressure at Assirik. If nesting has an anti-predator role, a reduction in predation pressure should lead to a reduction in nest height, and even terrestrial nesting [9]. This can be explained by recent data on large carnivore populations in NKNP, which show a sharp decline in the numbers of lions (*Panthera leo*), estimated at less than 50 individuals [17], and wild dogs (*Lycaon pictus*), which fell from around 400 individuals in 1975 [56] to 37 individuals in 2011 [18]. As for the leopard (*Panthera pardus*), which is probably the greatest threat to chimpanzees [57], the population is estimated at 403 individuals [18] and would be vulnerable [58]. Nevertheless, chimpanzee nests were located in hard-to-reach places, at the edge of gallery forests, at the ends of tree branches or near the crowns of trees, so as to detect the approach of predators or deter an attack. Further research is needed to evaluate this hypothesis.

Chimpanzees at Assirik were also selective in the choice of trees used for nest building. Of the 37 plant species recorded bearing nests, 12 were frequently used for nesting, which is far more than at most chimpanzee study sites in West Africa. In the Gola Rainforest National Park in Sierra Leone, chimpanzees nested in 22 tree species, and only six of these accounted for 70% of observations [42]. In Bagnomba, located 15 km from Niokolo Koba National Park, of the 26 or more tree species used for nesting, *Diospyros mespiliformis*, *Pterocarpus erinaceus* and *Anogeissus leiocarpus* accounted for more than 70% of nests [41]. Surveys by [14] showed that at Diaguiri, around 70 km from NKNP, nest construction was focused on five plant species: *Pterocarpus erinaceus*, *Anogeissus leiocarpus*, *Diospyros mespiliformis*, *Khaya senegalensis* and *Hexalobus monopetalus*. It is possible that, through its management, NKNP maintains a high level of undisturbed habitat that favors the use of a wide range of plant species for nesting.

The reliable assessment of ape population size from nest index surveys relies on accurate estimation of the local decay rate [53]. This decay rate varies considerably from one site to another, depending on rainfall, the physical characteristics of the supporting trees, the type of habitat, the nest height and even the pH [40,43,53,54,62]. In this study, the rate of nest decay obtained was relatively shorter than at other study sites in Africa: 112.8 days at Sapo (Liberia) [63], 111 days at Kibale (Uganda) [64], 91.22 days at Taï (Côte d’Ivoire) [40], 194 days in Haut Niger National Park (Guinea Conakry) [65], 176.5 days in gallery forests and 184 days in wooded areas in Diaguiri (Senegal) [14], and 127 days in Mbam-Djerem (Cameroon) [43]. The number and intensity of environmental factors influencing the rate of nest decay at each site could explain this variability. We found that the nest decay rate at Assirik was explained by season and the physical characteristics of the nest trees. Almost all of the nests marked at the beginning of the rainy season and a large part (90.2%) of those marked in mid-season deteriorated in a short time (60 days). Moreover, these rainy season nests rotted three times faster than those built in the dry season. The seasonal effect coefficients were negative, indicating that precipitation increases the nest decay rate. These results are similar to those of Kouakou et al. [40], who showed that rainfall increased the rate of nest degradation. A study on orangutan nesting ecology by [66] found that the nest decay rate depended on the type of tree used for construction. At Assirik, plant species such as *Hannoa undulata*, *Celtis intigrifolia* and *Pachystela pobeguiniana* had slower decay rates. Differences in decay rates among nesting tree species indicate that the physical properties specific to each tree species play a role in nest rotting. Although the height of the nests did not have a major impact on the rate of decay, the results revealed that the nests built higher up decayed at a slower rate, perhaps because they are more solidly built than nests at lower heights. We think that, because of the risk of fatal falls, chimpanzees build stronger, firmer nests at greater heights, which could slow down the rate of nest decay. Given that most nests were located in gallery forests, we could not test for an effect of habitat type on nest decay at Assirik, but [14] showed that the nest decay rate was faster in gallery forest, relative to woodland, at the Diaguiri site. Given the variability in nest decay rates across sites and the numerous factors that influence the decay rate even at sites that are in close proximity, it is crucial that researchers provide up-to-date local data for accurate nest index surveys of great ape populations [67].

Currently, actions to conserve the remaining populations of wild chimpanzees are ramping up. The execution of these initiatives fundamentally requires the availability of reliable and recent data. The scientific information emerging from this present study is of paramount importance for the preservation of natural habitats and floral resources of great interest for the conservation of chimpanzees in Senegal. Determining the rate of nest decomposition will not only make it possible in the future to estimate the density of chimpanzees at Assirik, a site where the last estimate dates back more than 20 years [7], but also to compare from up-to-date information, population demographic trends over time. We believe that the MNC method may be effective in determining the degradation rate of great ape nests, because it takes into account all of the factors that can influence nest decomposition and, more particularly, the effect of seasonal variations that are very important in savannah ecosystems. Furthermore, since most protected areas in Africa are often plagued by illegal activities, the MNC can be an effective means of monitoring the habitat quality of these great apes [39], in that repeat visits are a deterrent.

## 5. Conclusions

The results of our study show that chimpanzee nest building in Assirik primarily occurs in gallery forest habitat, and this pattern has been consistent over time. However, nest height has dropped considerably over a period of about 20 years (2000–2002: [9]; 2020–2022: this study), with nest height being strongly correlated with the height of nest trees (this study). Chimpanzees showed preferences for a wide range of plant species in the choice of trees to build their nests, including *Diospyros mespiliformis*, *Pterocarpus erinaceus*, *Malacantha alnifolia*, *Ficus sur*, *Celtis intigrifolia*, *Afzelia africana*, *Combretum collinum*, *Hexalobus monopetalus*, *Pachystela pobeguiniana*, *Hannoa undulata*, *Khaya senegalensis* and *Cola cordifolia*. The rate of nest decay in the study site is faster than in other chimpanzee study sites and is influenced by factors such as rainfall, tree species used for nesting, and nest height. In the light of these findings, it is important for researchers to provide up-to-date local information on chimpanzee nesting behavior for the proper management of this species.

## Figures and Tables

**Figure 1 animals-14-00553-f001:**
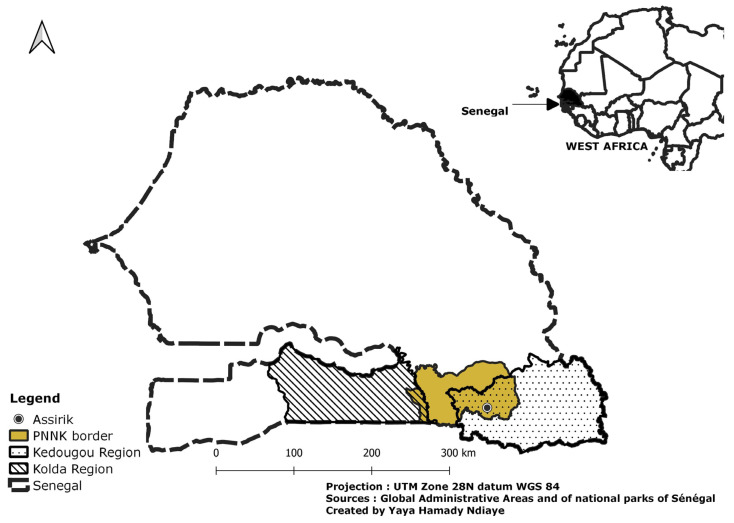
Location of the study area.

**Figure 2 animals-14-00553-f002:**
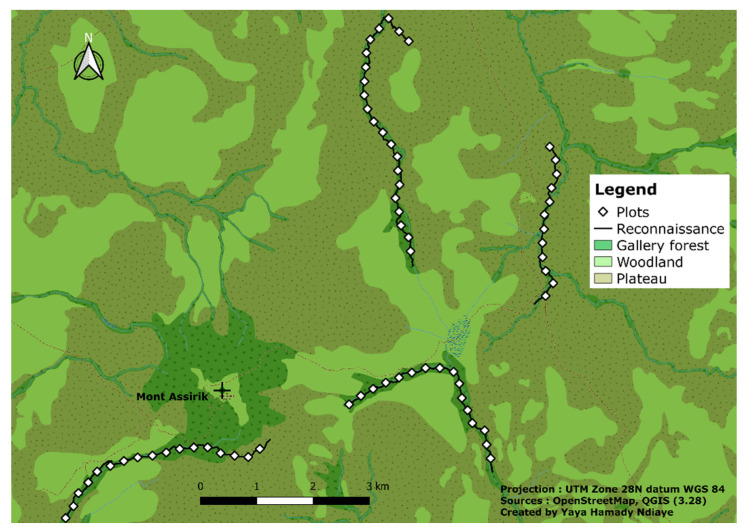
Reconnaissance path and botanical survey map.

**Figure 3 animals-14-00553-f003:**
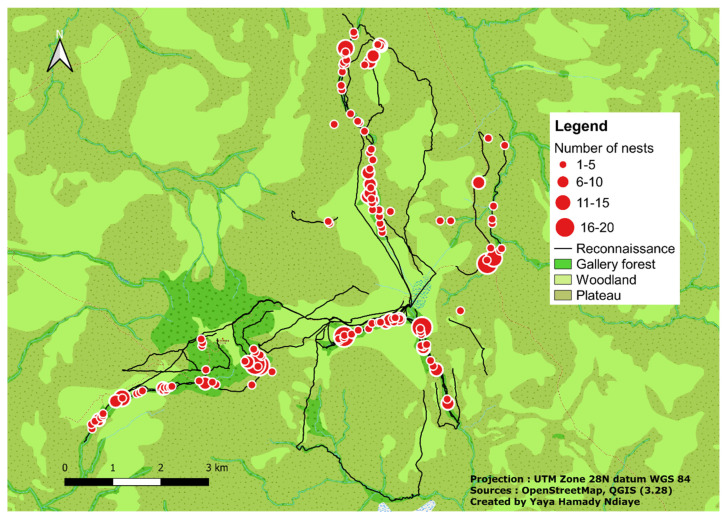
Distribution of chimpanzee nests in Assirik.

**Figure 4 animals-14-00553-f004:**
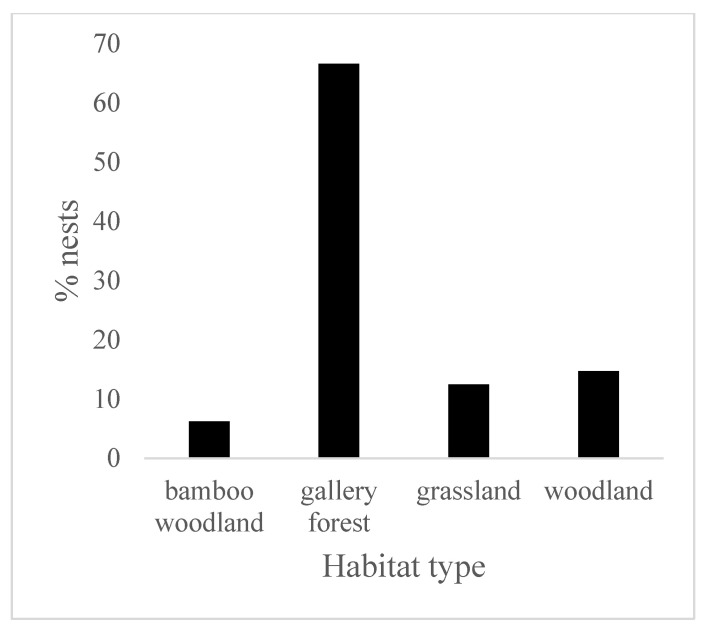
Percentage of nests (*n* = 626) by habitat type at Assirik.

**Figure 5 animals-14-00553-f005:**
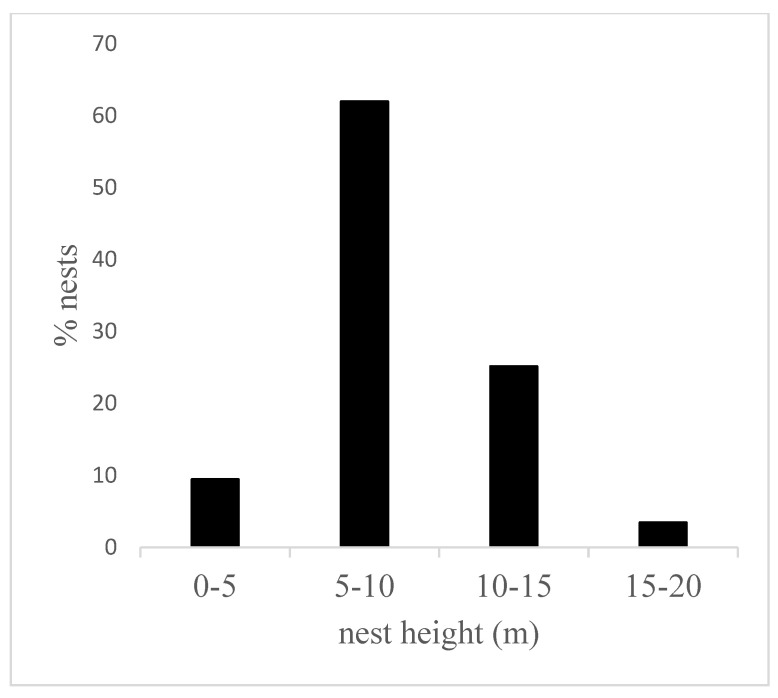
Percentages of nest height (*n* = 318) divided into four classes.

**Figure 6 animals-14-00553-f006:**
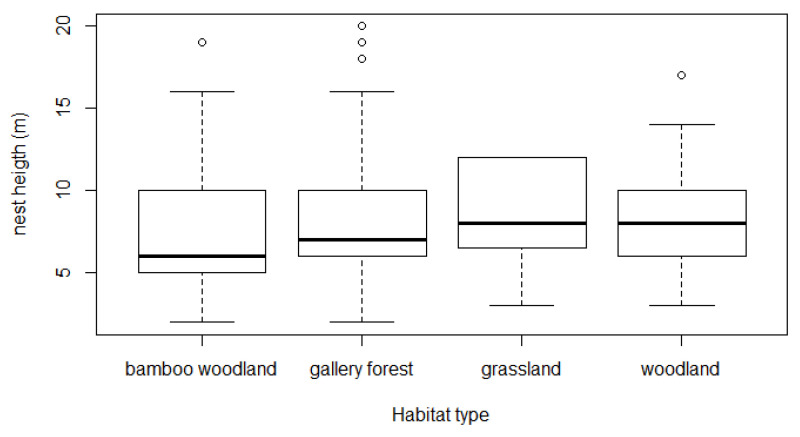
Graph of nest height as a function of habitat type (*n* = 318).

**Figure 7 animals-14-00553-f007:**
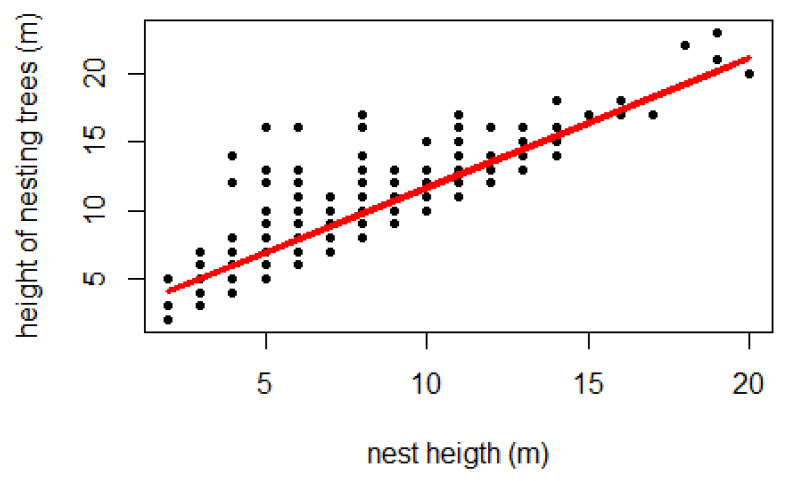
Linear regression line between nest height (*n* = 318) and support tree height (y = 0.94481x + 2.20008, r^2^ = 0.71).

**Figure 8 animals-14-00553-f008:**
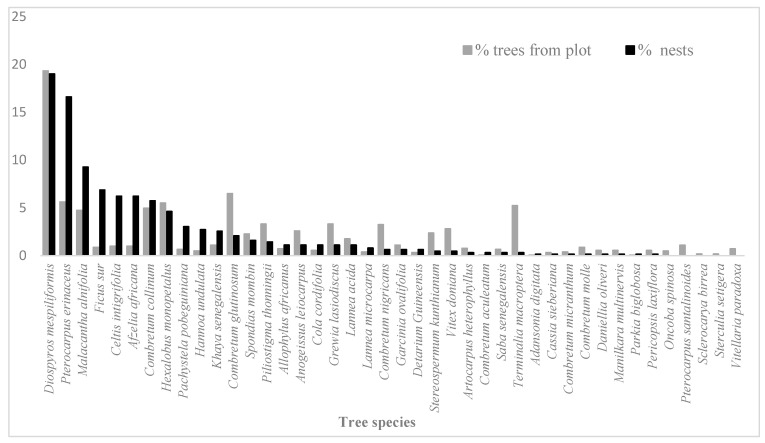
Percentages of trees (*n* = 1815) from plots and nesting trees (*n* = 626) used by chimpanzees.

**Figure 9 animals-14-00553-f009:**
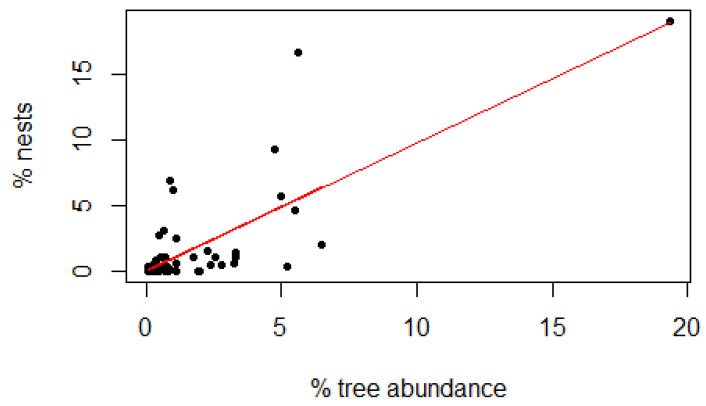
Relationship between trees used for nesting and their abundance at Assirik (y = 0.97378x + 0.04023, r^2^ = 0.5713).

**Figure 10 animals-14-00553-f010:**
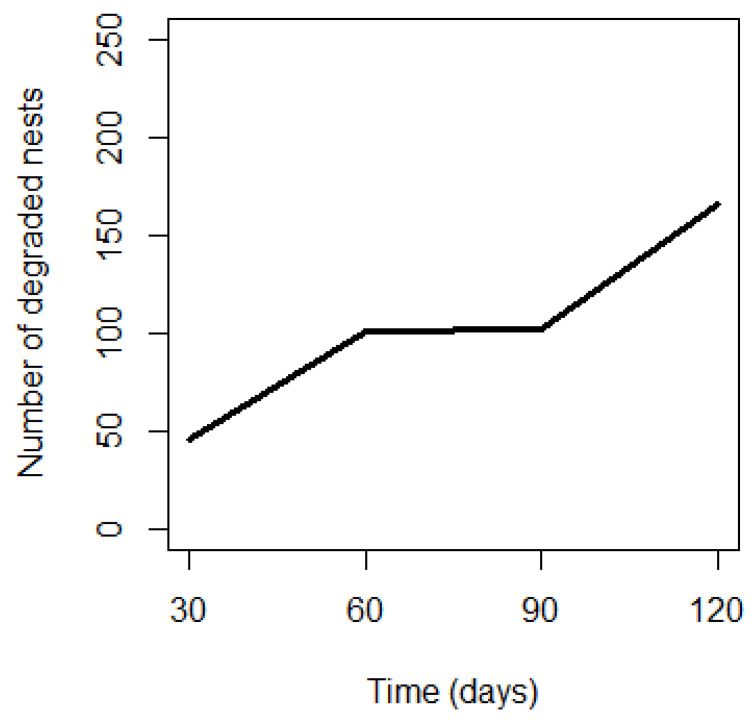
Nest degradation curve (*n* = 166) as a function of time.

**Table 1 animals-14-00553-t001:** Distribution of nests in the different types of vegetation according to the seasons.

Season	Dry Season	Wet Season	Total
Bamboo woodland	2	1	3
Gallery forest	115	54	169
Grassland	7	28	35
Woodland	12	22	34
Total	136	105	241

**Table 2 animals-14-00553-t002:** Average lifespan of nests according to tree species.

Nest Trees	Mean Days (sd)	Coefficient (Univariable)
*Afzelia Africana*	31.6 (5.1)	-
*Celtis intigrifolia*	95.3 (22.7)	10.00 (0.98 to 19.01, *p* < 0.001)
*Combretum collinum*	48.9 (11.6)	8.26 (1.31 to 15.21, *p* = 0.013)
*Diospyros mespiliformis*	65.4 (23.9)	15.04 (7.28 to 22.80, *p* < 0.001)
*Ficus sur*	87.3 (24.5)	6.78 (−1.72 to 15.27, *p* < 0.001)
*Grewia lasiodiscus*	48.0 (0.0)	16.11 (3.68 to 28.54, *p* = 0.199)
*Hannoa undulata*	109.0 (0.0)	10.05 (0.45 to 19.66, *p* < 0.001)
*Khaya senegalensis*	38.6 (14.6)	0.98 (−6.53 to 8.49, *p* = 0.339)
*Lannea microcarpa*	30.0 (0.0)	−2.07 (−10.36 to 6.21, *p* = 0.850)
*Malacantha alnifolia*	96.1 (20.8)	6.68 (−2.51 to 15.87, *p* < 0.001)
*Pachystela pobeguiniana*	103.2 (15.6)	7.30 (−2.75 to 17.36, *p* < 0.001)
*Pterocarpus erinaceus*	36.6 (9.1)	5.02 (−3.17 to 13.20, *p* = 0.552)
*Saba senegalensis*	45.0 (0.0)	14.64 (1.81 to 27.47, *p* = 0.294)
*Stereospermum kunthianum*	45.0 (21.2)	1.45 (−11.50 to 14.39, *p* = 0.294)

**Table 3 animals-14-00553-t003:** Average lifespan of nests according to group.

Dependent: Lifespan		Mean Days (sd)	Coefficient (Univariable)
Group	1	107.6 (4.6)	-
	2	64.1 (14.3)	−43.49 (−47.11 to −39.87, *p* < 0.001)
	3	34.9 (7.9)	−72.68 (−75.89 to −69.48, *p* < 0.001)

**Table 4 animals-14-00553-t004:** Average heights of chimpanzee nests at study sites in Senegal.

Study Site	Nest Height		
	Mean (m)	SD	References
Antenna zone	8.7	3.34	[36]
Bagnonba	7.90	3.62	[40]
Bantankiline	9.91	-	[44]
Diaguiri	10.9	1.7	[14]
Fongoli	8.33	4.13	[9]
Assirik	13.55	4.24	[9]
Assirik	8.07	0.36	This study

**Table 5 animals-14-00553-t005:** Average nest heights at study sites in Africa where chimpanzees coexist with large numbers of large carnivores.

Study Site	Nest Height		
	Mean (m)	SD	References
Issa	12.15	4.19	[59]
Sapo	10.85	8.3	[60]
Semliki	11	5.81	[47]
Taï	23.2	-	[61]
Assirik	8.07	0.36	This study

## Data Availability

All data are available upon request. Our data will be stored in the Purdue University library and in the Apes database after publication.

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
