# Peer review of "Updating Chimpanzee Nesting Data at Mount Assirik (Niokolo Koba National Park, Senegal): Implications for Conservation"

_animals, 2024, doi:10.3390/ani14040553_

Round 1

Reviewer 1 Report

Comments and Suggestions for Authors

This manuscript on nesting data of chimpanzees at Mount Assirik in Niokolo Koba National Park, Senegal was well written with important implications for understanding the ecology and population status in an area where previous data was lacking. A description of data collection and analysis was detailed and figures were well done to illustrate the results. 

One topic was missing from the introduction - the reviewer recommends that the authors add a short description of the importance of nests, their function in chimpanzee ecology (sleeping, predator evasion, use in reproduction - infant rearing, etc.),  and also how long the chimps use one nest (remain for days, new nest each night, etc.). Do males or females (or both) construct nests? Do they construct them at different heights? This will help in understanding the importance of new nest construction and decay rates and how the presence of new nests can indicate population status.

There are a few minor edits (noted in the attached manuscript), but otherwise the manuscript was well done!

Author Response

We uploaded our responses to reviewers in the attached file.

Best regards

Reviewer 2 Report

Comments and Suggestions for Authors

This is basically an excellent manuscript that describes a very good study.  Its justification is well established in the Introduction.  The data is very well presented and analyzed.  And the Discussion section very nicely couches the findings in an appropriate context.  Overall, it was a pleasure to read. 

Its major drawback is that this study is admittedly quite narrow in focus in that it reports only on the number of nests observed in only a limited area within one particular park in Senegal.  Nevertheless, the data provided will provide a valuable update relevant to the management in that particular park.  And of more general interest, this paper will serve as an exemplary model of thorough reporting of this type of information. 

Upon review, I have only the following three questions/concerns:

That nest-decay rate can affect the accuracy of population estimates should be explained more clearly in the Introduction.  Can an elaboration of the point made on Line 366 be developed in the Intro?

Line 128:  “episodically” needs to be explained in greater detail.  How many specific survey days occurred, and at what times of the year? 

Lines 366-396:  Could the population size of the chimpanzees in the study area be estimated?  If not, why not?

In addition, and only in an effort to be helpful, I offer the following minor suggestions/corrections: 

Lines 125-126:  That Excel was used to tabulate the data is irrelevant.  The data was collected and tabulated.  That matters.  The specific brand of spreadsheet does not.  Likewise, the particulars about the statistical tests used does matter.  That they were written in R, or any other code, does not. 

Line 138:  “classification of [52]”  should be “classification of Tutin and Fernandez [52]”

Line 170:  I believe by “future” they mean “subsequent.”

Line 173:  That a t test was used is important.  That is was coded in R is irrelevant. 

Line 207:  This sentence is unnecessary, and should be dropped.  It provides an incomplete version of the more clear information in the next paragraph. 

Line 330:  “[9] obtained….”  should read “Pruetz et al [9] obtained….”

Line 354:  This is an incomplete comparison.  Less selective than what?  Than in other locations, is what I believe should be added to this sentence. 

Line 366:  The importance/value of decay rate should also be explained in the Introduction of this manuscript. 

Line 381:  Same problem described for Lines 138 and 330. 

These items notwithstanding, I repeat that I find this to be a very good paper, one that I hope will be published.  I offer these authors my congratulations for bringing this project to fruition. 

Author Response

Dear

We uploaded our responses to reviewers in the attached file.

Best regards
